# Hidden threats in urban environments: *Angiostrongylus cantonensis* in Banda Aceh's cityscape

**Lucia Anettová**[1]*, **Anna Šipková**[2], **Vojtech Baláž**[3], **Muhammad Hambal**[4], **Radovan Coufal**[2], **Jana Kačmaříková**[5], **Henni Vanda**[4], **Wahyu Eka Sari**[4], **David Modrý**[1,2,6]

1 Department of Veterinary Sciences, Faculty of Agrobiology, Food and Natural Resources, Czech University of Life Sciences, Prague, Czech Republic, 2 Department of Botany and Zoology, Faculty of Science, Masaryk University, Brno, Czech Republic, 3 Department of Ecology and Diseases of Zoo Animals, Game, Fish and Bees, Faculty of Veterinary Hygiene and Ecology, University of Veterinary Sciences, Brno, Czech Republic, 4 Faculty of Veterinary Medicine, University Syiah Kuala, Banda Aceh, Indonesia, 5 Department of Pathology and Parasitology, Faculty of Veterinary Medicine, University of Veterinary Sciences, Brno, Czech Republic, 6 Institute of Parasitology, Biology Center, Czech Academy of Sciences, České Budějovice, Czech Republic

☉ These authors contributed equally to this work.

* lucia.anettova@gmail.com

**Data availability statement:** All data generated or analyzed during this study are included in this published article.

## Abstract

*Angiostrongylus cantonensis* is a mollusk-borne parasitic nematode originating in Southeast Asia. Commonly known as the rat lungworm, it uses rats as definitive hosts, though other mammals, including humans, can be infected and typically suffer from neurological disorders. This study focuses on the parasite's presence in its gastropod intermediate hosts in several urban and rural areas in Aceh province, Sumatra, Indonesia. Samples of *Achatina* (*Lissachatina*) *fulica* (161) and *Pomacea* sp. (90) were collected in eight localities in Banda Aceh. Additionally, 531 edible freshwater snails belonging to the genus *Sulcospira* sp. from three different localities in Aceh province were obtained in wet markets. All samples were examined by LAMP and qPCR for the *A. cantonensis* DNA. No samples of *Sulcospira* sp. and *Pomacea* sp. tested positive. 13.4% of *L. fulica* tested positive, with the highest prevalence in urban areas of Banda Aceh. The ITS1 sequences obtained from positive samples using conventional PCR confirmed 100% identity with *A. cantonensis*. The present study confirms, for the first time, the presence of the zoonotic parasite *A. cantonensis* in Banda Aceh, Sumatra, Indonesia. Notably, the handling and consumption of snails sold at wet markets do not appear to increase the risk of eosinophilic meningitis in this region. However, the relatively high prevalence of *A. cantonensis* in urban land snails underscores the need for continued vigilance and public health awareness.

**Funding:** This work was supported by the SEAEUROPEJFS19IN-053 project and the Czech Science Foundation grant no. 22-26136S (both to DM). The work of Lucia Anettová (LA) and Anna Šipková (AŠ) was also supported by Specific Research – support of student projects, no. MUNI/A/1602/2023. The funders had no role in study design, data collection and analysis, decision to publish, or preparation of the manuscript.

**Competing interests:** The authors have declared that no competing interests exist.

## Author summary

*Angiostrongylus cantonensis*, commonly known as rat lungworm, is a parasitic nematode that can cause serious neurological disease in humans and other accidental hosts. It is transmitted via snail or slug intermediate hosts, typically through accidental ingestion of infected snails or contaminated produce. In this study, we investigated the presence of *A. cantonensis* in land and freshwater snails from urban and rural areas of Aceh province, Sumatra, Indonesia — region where its occurrence had not previously been documented. We tested over 780 snails using molecular diagnostic techniques and found that *Achatina* (*Lissachatina*) *fulica*, an invasive land snail commonly found in urban environments, had a relatively high infection rate. In contrast, none of the freshwater snails sold for consumption at local wet markets tested positive. Our findings represent the first molecular confirmation of *A. cantonensis* in Banda Aceh. While current practices around the handling and consumption of freshwater snails may pose little risk in this area, the presence of the parasite in urban land snails highlights the importance of public health awareness and monitoring, especially in densely populated settings where contact with infected snails is more likely.

## Introduction

The rat lungworm *Angiostrongylus cantonensis* is a food-borne zoonotic metastrongyloid nematode first described in Canton, China [1]. However, it has expanded throughout the world and is considered an emerging pathogen [2], causing eosinophilic meningitis in humans and similar neurological disorders in numerous mammalian and avian species considered aberrant hosts [3,4].

Cases of human eosinophilic meningitis likely caused by *A. cantonensis* have been reported from Southeast Asia since the 1960s [5,6]. The parasite's presence in both, definitive and intermediate hosts or in humans (as its accidental hosts) have been confirmed on most of the main islands of Indonesia [6–9]. In Sumatra, eight human cases of eosinophilic meningitis likely caused by *A. cantonensis*, were first reported in Kisaran, North Sumatra Province [10]. The parasite was also identified in rats (*Rattus tiomanicus jalorensis* and *Rattus rattus diardii*), originating from the surroundings of Medan, North Sumatra Province [11]. Although no further human infections have been documented, the parasite has been detected in rats and snails (including *A. fulica*) across the island, particularly in South Sumatra Provinces (Lubuk Linggau, Baturaja, and Martapura) and Lampung Province [7,12,13]. However, the northwestern part of Sumatra remains largely unstudied.

The giant terrestrial African snail (*A. fulica*, Fig 1A) and the freshwater apple snail (*Pomacea* spp., Fig 1B) are among notable invasive mollusks associated with the zoonotic transmission of *A. cantonensis* to humans [2,14,15]. Resulting from a successful invasion, the abundance of these gastropods in Aceh province is high [16,17] and these species could serve as suitable terrestrial and aquatic intermediate hosts for *A. cantonensis* in the area. Although widely distributed in Aceh province, these mollusks are generally not used for human consumption, in contrast to freshwater *Sulcospira* spp. (Fig 1C) commonly consumed by the local population. However, the potential role of this species as an intermediate host for *A. cantonensis* remains undetermined.

Rats of the genus *Rattus* are definitive hosts for *A. cantonensis*, with gastropods serving as intermediate hosts. The parasite's life cycle involves first-stage larvae (L1) excreted in rat feces, developing into infective third-stage larvae (L3) within gastropods [18]. Poikilothermic

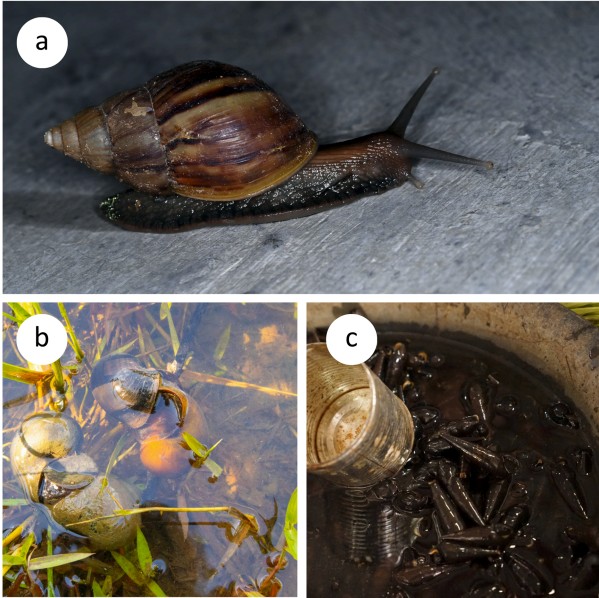

**Fig 1. Species of gastropods examined for the presence of *Angiostrongylus cantonensis* larvae in Aceh province, Indonesia (A) Invasive landsnail *Achatina* (*Lissachatina*) *fulica* found in the campus of the University Syiah Kuala, Banda Aceh; (B) Invasive aquatic snail *Pomacea* sp. found in a pond of the rural area near Banda Aceh; (C) Edible aquatic snail *Sulcospira* sp. from wet markets, originally from extensive snail farms in Aceh province.** Photographs taken by RC and DM.

animals (including reptiles and amphibians) can act as paratenic hosts [19–22], transmitting infection to humans and other mammals. Due to the invasion and spread of the parasite, facilitated by rats and gastropods, resulting disease is becoming increasingly common in humans and is considered one of the emerging infectious diseases [2].

Apart from direct ingestion of intermediate or paratenic hosts, food, water or hands contaminated with free L3 released from gastropods represent another route of transmission to humans [18,23]. In the human body, the parasites reach the brain in the same way as in rats and develop to the subadult stage. The immune reaction and the presence of larvae in the brain result in severe neurological symptoms, occasionally with fatal consequences [24].

Aceh Province is the northernmost part of Sumatra, characterized by a warm and humid climate that is conducive to the life cycle of *A. cantonensis*. Culinary habits in Aceh include the consumption of *Sulcospira* snails as well as other freshwater invertebrates (such as crustaceans) and vertebrates (e.g., fish), which are potential paratenic hosts for *A. cantonensis* [22], underscoring the need for ongoing monitoring and public health vigilance. Our study investigates the presence of *A. cantonensis* in intermediate hosts in urban, suburban and rural areas of Banda Aceh and potential risk for zoonotic infections. To the authors´ knowledge, this is the first study to investigate the presence of *A. cantonensis* in Aceh province.

## Materials and methods

### Sample collection

Landsnails *A. fulica* were collected in five localities, two urban: city center (1) and campus of the University Syiah Kuala (2); one suburban: Lambirah (3); and two rural: pastures near Kuede (4) and gardens by Nurul Qualbi (5). The urban locality in the city center was further

divided into two sublocalities: park (1a) and riverside (1b) (Table 1, Fig 2). The localities were characterized as urban, suburban or rural based on land use and infrastructure in the particular areas, taking into account population density in the areas. In each locality/sublocality, at least 30 snails were collected, except for the locality 4, where we were able to find only 11 individuals. All collected *A. fulica* were euthanized by freezing.

*Pomacea* sp. were collected in three localities: two suburban (Lambirah and along Jl. Tgk. Bakurma) and one urban (University Syiah Kuala campus). Thirty snails were collected from each locality. A different methodology was applied for *Pomacea* sp. due to their extremely hard shells and tight-closing operculum, which make standard dissection impractical. Snails were crushed in sealed plastic bags filled with water (approx. 50 ml), allowing larvae to emerge and sediment for downstream molecular analysis, following the approach of Modrý et al. [25]. After one hour, 0.5 ml of sediment was collected from each bag for DNA extraction using PrepMan, to be used in the LAMP assay.

In wet markets, edible freshwater snails *Sulcospira* sp. from three different localities in Aceh province were obtained. 150 snails were obtained from Lhoong, Aceh Besar District, 91 from Leupung Subdistrict, Aceh Besar District and 290 from Lamno, Aceh Jaya District. The

**Table 1. Sampling localities of *Achatina* (*Lissachatina*) *fulica* in Banda Aceh and its surroundings, with the categorization and numbers of samples tested positive by LAMP and qPCR.**

| Locality | Type | No. of Samples | No. Positive |
|---|---|---|---|
| (1a) City center (park) | urban | 32 | 6 |
| (1b) City center (riverside) | urban | 31 | 11 |
| (2) University Syiah Kuala | urban | 30 | 3 |
| (3) Lambirah | suburban | 30 | 0 |
| (4) Kuede | rural | 11 | 0 |
| (5) Nurul Qualbi gardens | rural | 30 | 2 |

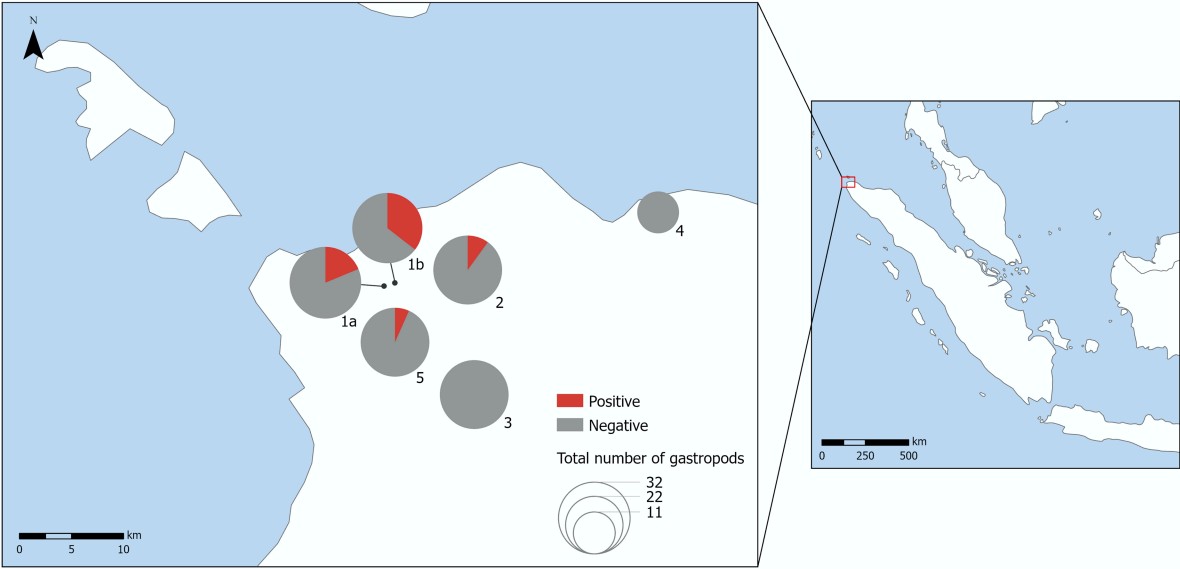

**Fig 2. Sampling localities of *Achatina* (*Lissachatina*) *fulica* for the detection of *Angiostrongylus cantonensis* DNA in Banda Aceh, Indonesia. The map was created by the authors using ArcGIS and base layers from Natural Earth (http://www.naturalearthdata.com).** All base layers are in the public domain and freely available under the terms of use specified at: http://www.naturalearthdata.com/about/terms-of-use/.

snails from wet markets were treated the same way as *A. fulica*. The snails were identified to the genus level based on Köhler et al. [26].

Both aquatic snails were identified morphologically to the genus level, due to the limited availability of studies on aquatic gastropods in Indonesia. The taxonomy of the genus *Sulcospira* in Sumatra remains unresolved, and the exact species present in the region are currently unknown. Specimens of *Pomacea* likely belong to either *P. canaliculata* or *P. maculata*, two morphologically similar species that can be reliably distinguished only through genetic analysis [27,28]. In contrast, *A. fulica* is morphologically distinct and easily identifiable to the species level. All the gastropod species collected are shown in Fig 1.

## Molecular analysis and sequence data

**LAMP.** All samples were initially screened using the LAMP assay due to its lower cost and simpler workflow compared to qPCR, allowing for rapid elimination of negative pools and more efficient processing. In case of *A. fulica*, pools of three individuals were used, using approximately 10 μg of a foot tissue from each snail. Tissue samples from each individual were also stored for further qPCR analysis in case the pooled LAMP analysis was positive. First, DNA was extracted from snail tissues using PrepMan Ultra Sample Preparation Reagent (Thermo Fisher Scientific, Waltham, MA, USA), according to the manufacturer's instructions. Briefly, PrepMan and tissue were used in a 1:1 ratio in 2mL vials, i.e., 50–150 $\mu$ L of PrepMan was added depending on the tissue amount. The mixture was incubated at 99 °C in a water bath for 10 minutes and used immediately. The LAMP assays were performed using the Genie II platform (OptiGene, UK) in 20 $\mu$ L reactions, as described by Baláž et al. [29]. In case of *Pomacea* sp. and *Sulcospira* sp., pools of three and 10 individuals respectively were processed by the LAMP in the same way as the samples of *A. fulica*. Different pool sizes were used, depending on the size of the snails.

**qPCR.** Pooled samples which tested positive with LAMP subsequently underwent the species specific qPCR. DNA from all the individuals in pools that tested positive by LAMP was subsequently extracted using DNEasy Blood&Tissue (Qiagen, Germany), extending the pre-lyse phase overnight as a modification for L3 of *A. cantonensis*. Approximately 25 $\mu$ g of foot tissue was used. The qPCR assay was performed on BIO-RAD CFX96 Real Time system in a 20 $\mu$ L reaction, according to Sears et al. [30] and modified as in Anettová et al. [31]. As positive controls, DNA from a single L3 of *A. cantonensis* extracted by the same method as samples and diluted 100x was used.

Additionally, SYBR Green-based quantitative real-time PCR as in Jakkul et al. [32] was performed on the DNA samples tested positive previously on qPCR to investigate possible co-occurence of *A. malaysiensis*. The assay was run in duplicates and two sets of primers for the two nematodes were used. In case of confirmation of *A. cantonensis* the primers were used as originally published. In case of *A. malaysiensis*, we have adjusted the length of primers, to cover more variable positions and increase annealing temperature. The primers used in SYBR Green qPCR can be found in Table 2.

**Conventional PCR.** To confirm the identity of larvae as *A. cantonensis* in *A. fulica*, nested PCR was performed to obtain ITS1 sequences from the snail individuals that tested positive on qPCR. This involved using the ITS1-Canto-F3 (5′AAC AAC TAG CAT CAT CTA CGTC 3′) and ITS1-Canto-R1 (5′CAT CCT GTG TAT CTC GTT CC 3′) primers targeting the ITS-1 region of *A. cantonensis*, following the PCR cycling conditions: 95°C for 15 min, followed by 35 cycles of 94°C for 30 s, 57°C for 90 s and 72°C for 90 s, with a final step of 72°C for 10 min. The reaction was performed in a total volume of 25 $\mu$ L, containing 2 $\mu$ M of each primer,

**Table 2. Primers used in SYBR Green qPCR according to Jakkul et al. [32], with primers for *A. malaysiensis* modified.**

|  | *A. cantonensis* | *A. malaysiensis* |
|---|---|---|
| Forward | AC4_cytb_F<br>(5′ AAT GTT TGT TGA GGC AGA TC 3′) | AM_cytb_Fnew<br>(5′ CGA GAT ATT TAT TGA GGC TGA TCC TTT GAC 3′) |
| Reverse | AC5_cytb_R<br>(5′ GCT ACA ACA CCC ATA ACC T 3′) | AM_cytb_Rnew<br>(5′ GAC AAA ACC CTC ATC AAT AAA GCC A 3′) |

12.5 $\mu$L of Multiplex PCR Master Mix and $\mu$L of DNA template., as described by Izquierdo-Rodriguez et al. [33], from which these primers and conditions were originally adopted. PCR products were purified with ExoSAP-IT (Affymetrix, ThermoFisher Scientific, Czech Republic) and sequenced commercially at Macrogen Europe (Amsterdam, Netherlands) using the amplification primers. Sequences were assessed and trimmed manually in Geneious Prime 2024.0.5 (http://www.geneious.com) and checked against BLAST [34].

Prevalence was calculated as the proportion of positive samples among the total number examined, and 95% confidence intervals were computed using the Clopper–Pearson exact method. The analysis was descriptive and focused on summarizing prevalence across species and localities.

## Results

Of the 55 pools of *A. fulica*, 12 tested positive. All pools that were positive by LAMP (each consisting of three *A. fulica* individuals) were confirmed by qPCR, with at least one individual in each pool testing positive. The terrestrial snail *A. fulica* was the only species that tested positive for *A. cantonensis* DNA on both, LAMP and qPCR. Of the 164 specimens collected in Banda Aceh municipality and its surroundings, 22 (13.4%, 95% CI: 8.6–19.4%) tested positive. The highest prevalence was observed in the city center sublocalities: 6 of 32 samples (18.8%; sublocality 1a) and 11 of 31 samples (35.5%; sublocality 1b). At campus of University Syiah Kuala, 3 of 30 samples (10%) were positive, with one additional sample yielding dubious result. Among rural localities, only Nurul Qualbi (Loc 5) showed positive results, with 2 of 30 samples (6.7%) positive. No positive samples were found in Lambirah and Kuede. Aquatic snail *Sulcospira* sp. tested negative for *A. cantonensis* DNA in all examined localities by both assays, LAMP and qPCR. The results of *A. fulica* examination are summarized in Table 1; examples of positive and negative localities are shown in Fig 2. Since none of the aquatic snail pools tested positive by LAMP, no further analysis by qPCR was performed.

Five DNA sequences of *A. cantonensis* (BLAST analysis showed 100% identity with GenBank sequences of *A. cantonensis*) recovered from *A. fulica* samples were obtained from two different localities (City Center - park and Nurul Qualbi gardens) varying from 322 to 554 bp. The sequences obtained were identical, and one representative sequence from each locality has been submitted to GenBank with accession numbers PV133028 and PV133029. All DNA samples analyzed using SYBR Green-based real-time PCR tested positive with primers specific for *A. cantonensis*, while none tested positive for *A. malaysiensis*.

## Discussion

The study provides further data on distribution of *A. cantonensis* in Sumatra [10,11], with first data from this region. Our findings confirm the presence of *A. cantonensis* in the municipality of Banda Aceh, with DNA detected only in the terrestrial African giant snail

*A. fulica* with a remarkable prevalence, especially in the city center. Conversely, all examined samples from aquatic gastropods tested negative, despite a relatively large sample size of edible aquatic snails, including *Sulcospira* sp. and *Pomacea* sp. These results suggest absence or a very low presence of *A. cantonensis* in the aquatic environment in tested localities of the Aceh province.

To confirm the taxonomic identity of the nematode and to exclude possible co-occurence with *A. malaysiensis* as *A. cantonensis*, ITS1 sequence were analyzed and the SYBR-Green based real-time PCR was conducted, providing an ultimate taxonomic identification. While the qPCR assay used is highly sensitive and specific [30], it is not suitable for distinguishing among the three sibling species: *A. cantonensis*, *A. malaysiensis* (which likely overlaps with *A. cantonensis* in Southeast Asia), and *A. mackerrasae* (found in Australia).

The absence of *A. cantonensis* in *Pomacea* sp. in areas where *L. fulica* tested positive was unexpected. This disparity may indicate low exposure of aquatic mollusks in tested localities to *A. cantonensis* larvae or that *Pomacea* sp. are less suitable intermediate hosts compared to *A. fulica*. Previous studies support the latter hypothesis, showing lower prevalence in *Pomacea* spp. compared to terrestrial gastropods such as *A. fulica* [35]. However, no experimental studies have directly compared the suitability of these two intermediate hosts for supporting *A. cantonensis* larvae, highlighting a potential area for future research. Alternatively, this absence in aquatic hosts could also indicate an early stage of introduction in Aceh or environmental factors that limit the parasite's spread and development in these hosts and habitats. Further studies are needed to explore these factors and their influence on host-parasite dynamics.

*Sulcospira* sp., an edible aquatic snail frequently consumed as a traditional protein source in Aceh, tested negative for *A. cantonensis* DNA. To the authors' knowledge, this species has not previously been confirmed as an intermediate host. Several factors may explain the absence of *A. cantonensis* in the examined samples. First, all edible aquatic snails analyzed in this study originated from areas outside the Banda Aceh municipality, in contrast to the urban land snails (*A. fulica*) that tested positive. Second, although *A. cantonensis* is known for its low host specificity and ability to infect a wide range of gastropod species [18], it remains possible that *Sulcospira* sp. has a lower susceptibility or is less suitable as an intermediate host. Third, the aquatic snails were sourced exclusively from three extensive snail farms, which may represent simplified ecological environments with limited exposure to the full parasite life cycle — particularly if rat populations, essential for maintaining transmission, are sparse or absent.

Despite the negative results, the potential role of edible aquatic snails in transmission should not be underestimated. Handling and processing practices may still pose a risk of infection, especially if snails are undercooked or contaminated during transport or preparation. These findings underscore the importance of further investigations into aquatic hosts of *A. cantonensis* in the region, particularly under natural or semi-natural conditions, to better understand the transmission dynamics and potential risks to public health.

Our results emphasize predominant association of *A. cantonensis* with urban areas and its much lower prevalence in suburban and rural regions. This pattern may be driven by higher population densities of *A. fulica* in urban environments [36,37], as this highly invasive species has a lower chance to be introduced directly into the natural or rural environment. However, as the snails and the parasite continue to spread, they are likely to eventually colonize these areas and find the environment suitable. Demonstrated presence in urban environment underscores risk of possible transmission via contaminated food or water.

The presence of *A. cantonensis* in this newly recognized area highlights the need for public health interventions, such as minimizing human contact with terrestrial snails, particularly in urban settings. Given the frequent consumption of freshwater gastropods and other

aquatic hosts in Aceh, the potential for infection via aquatic pathways cannot be ignored. Further research should explore the prevalence of *A. cantonensis* in these hosts to assess the risk posed by local dietary practice [38,39]. Neuroangiostrongyliasis remains a significant public health concern and increased awareness is essential to mitigate its impact. Finally, the existence of underdiagnosed cases of eosinophilic meningitis in the region, particularly in areas with limited healthcare resources, should be considered [2,40].

## Conclusion

This study provides the first molecular evidence of *A. cantonensis* in Banda Aceh, Sumatra, with high prevalence detected in *A. fulica*, particularly in urban areas. The absence of the parasite in examined aquatic gastropods, including *Pomacea* sp. and *Sulcospira* sp., suggests either low environmental exposure or limited suitability of these species as intermediate hosts. The findings underscore the potential public health risk in urban settings and highlight the need for increased surveillance, public education, and further research on alternative transmission pathways and host-parasite dynamics in both terrestrial and aquatic environments.

## Acknowledgments

The authors thank Petr Janoš for creating the map of localities studied for the presence of *Angiostrongylus cantonensis* in *Achatina* (*Lissachatina*) *fulica* (Fig 2).

## Author contributions

**Conceptualization:** Anna Šipková, Muhammad Hambal, David Modrý.

**Data curation:** Lucia Anettová, Anna Šipková.

**Formal analysis:** Vojtech Baláž.

**Funding acquisition:** Lucia Anettová, Muhammad Hambal, David Modrý.

**Investigation:** Lucia Anettová, Anna Šipková, Vojtech Baláž, Radovan Coufal, Jana Kačmaříková.

**Methodology:** Lucia Anettová, Anna Šipková, Vojtech Baláž, Radovan Coufal, David Modrý.

**Project administration:** Muhammad Hambal, Henni Vanda, Wahyu Eka Sari, David Modrý.

**Resources:** Muhammad Hambal.

**Supervision:** David Modrý.

**Validation:** David Modrý.

**Visualization:** Anna Šipková, Radovan Coufal.

**Writing – original draft:** Lucia Anettová, Anna Šipková.

**Writing – review & editing:** Lucia Anettová, Anna Šipková, Vojtech Baláž, Muhammad Hambal, Radovan Coufal, David Modrý.

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
