## [Decision Letter · Decision Letter 0]

22 May 2025

PNTD-D-25-00434

Hidden Threats in Urban Environments: Angiostrongylus cantonensis in Banda Aceh’s Cityscape

Dear Dr. Anettová,

Thank you for submitting your manuscript to PLOS Neglected Tropical Diseases. After careful consideration, we feel that it has merit but does not fully meet PLOS Neglected Tropical Diseases's publication criteria as it currently stands. Therefore, we invite you to submit a revised version of the manuscript that addresses the points raised during the review process.

Please submit your revised manuscript within 60 days Jul 21 2025 11:59PM. If you will need more time than this to complete your revisions, please reply to this message or contact the journal office at plosntds@plos.org. Please include the following items when submitting your revised manuscript:

We look forward to receiving your revised manuscript.

Kind regards,

Alessandra Morassutti, PhD

Academic Editor

Francesca Tamarozzi

Section Editor

Shaden Kamhawi

co-Editor-in-Chief

Paul Brindley

co-Editor-in-Chief

**Journal Requirements:**

At this stage, the following Authors/Authors require contributions: Lucia Anettová, Anna Šipková, Vojtech Baláž, Muhammad Hambal, Radovan Coufal, Jana Kačmaříková, Henni Vanda, Wahyu Eka Sari, and David Modrý. Please ensure that the full contributions of each author are acknowledged in the "Add/Edit/Remove Authors" section of our submission form.

4) We do not publish any copyright or trademark symbols that usually accompany proprietary names, eg ©,  ®, or TM  (e.g. next to drug or reagent names). Therefore please remove all instances of trademark/copyright symbols throughout the text, including:

- ® on page: 3.

Potential Copyright Issues:

- Please confirm (a) that you are the photographer of Figures 1, 2, and & 3., or (b) provide written permission from the photographer to publish the photo(s) under our CC BY 4.0 license.

- Figure 4. Please (a) provide a direct link to the base layer of the map (i.e., the country or region border shape) and ensure this is also included in the figure legend; and (b) provide a link to the terms of use / license information for the base layer image or shapefile. We cannot publish proprietary or copyrighted maps (e.g. Google Maps, Mapquest) and the terms of use for your map base layer must be compatible with our CC BY 4.0 license.

6) Please amend your detailed Financial Disclosure statement. This is published with the article. It must therefore be completed in full sentences and contain the exact wording you wish to be published.Please ensure that the funders and grant numbers match between the Financial Disclosure field and the Funding Information tab in your submission form. Note that the funders must be provided in the same order in both places as well.

**Reviewers' Comments:**

Reviewer's Responses to Questions

**Key Review Criteria Required for Acceptance?**

**Methods:**

-Are the objectives of the study clearly articulated with a clear testable hypothesis stated?

-Is the study design appropriate to address the stated objectives?

-Is the population clearly described and appropriate for the hypothesis being tested?

-Is the sample size sufficient to ensure adequate power to address the hypothesis being tested?

-Were correct statistical analysis used to support conclusions?

-Are there concerns about ethical or regulatory requirements being met?

Reviewer #1: The sample size sufficient to ensure adequate.

I suggest that the authors detail the methodology used for each stage of this study, as it can be reproduced by other researchers.

Reviewer #2: Well written. But sample size is small. Can be accepted as short note

Pls provide the primer sequences of ITS1 with thermal cycle of the ITS1 based PCR

Reviewer #3: (No Response)

**Results**

-Does the analysis presented match the analysis plan?

-Are the results clearly and completely presented?

-Are the figures (Tables, Images) of sufficient quality for clarity?

Reviewer #1: In their results, they could include a comparison between the results of the LAMP and qPCR assays. Both were performed, if I understood correctly. Did any sample present different results between them? Please make this point clearer.

Reviewer #2: Clearly stated results.

Make a single figure with Fig 1, 2, 3

Replace Fig 2 and 3 with clearer images as seen in fig 1

Reviewer #3: (No Response)

**Conclusions**

-Are the conclusions supported by the data presented?

-Are the limitations of analysis clearly described?

-Do the authors discuss how these data can be helpful to advance our understanding of the topic under study?

-Is public health relevance addressed?

Reviewer #1: The authors conclude their discussion appropriately

Reviewer #2: Please add few lines to conclude the study based on your solid data.

Reviewer #3: (No Response)

**Editorial and Data Presentation Modifications?**

Reviewer #1: Below, I provide some suggestions and other comments will be in the manuscript.

Pg 1 line 16 - Lissachatina fulica – Lissachatina is a subgenus: Achatina (Lissachatina) fulica. There is not enough phylogenetic support to elevate it to a genus. Many authors have used it as a genus in recent years, which causes a lot of confusion. I suggest changing it throughout the text. The subgenus should be cited only at the beginning of the text. After that, use only “A. fulica”

Pg 1 Line 28 – abbreviate A. cantonensis after first mention in the text

Pg 1 Line 45 – Please cite examples of freshwater vertebrates

Pg 3 Line 61 – Please describe the LAMP assay to help the reader understand better. But only do so in the next topic. How was this pooled analysis performed? What steps were taken? And how much of each mollusk was analyzed?

In the topic “Sample collection (52-76)”, I suggest separating the information that actually deals with the collection and, only then, talking about the procedures for parasitological analysis of each group of mollusks. I believe it will be clearer this way.

The next topic deals with molecular analyses. In it, the authors can talk about the LAMP method.

Pg 3, Line 108 – I suggest starting with “The terrestrial snail A. fulica…”.

Pg 4, Lines 119-121: I suggest: Five DNA sequences of A. cantonensis recovered from A. fulica samples were obtained from two different localities (City Center - park and Nurul Qualbi gardens) ranging from 322 to 554 bp.

Reviewer #2: Minor revision

Reviewer #3: (No Response)

**Summary and General Comments**

Reviewer #1: The manuscript is original, it is well written and presents interesting data on the malacological surveillance of natural hosts of A. cantonensis and reinforces Achatina fulica as the intermediate host that most favors the maintenance of the parasite cycle in the environments studied. This has been demonstrated in several countries with a large dispersion of this snail, such as in Brazil. Some other references can be added. I suggest that the authors detail the methodology used for each stage of this study, as it can be reproduced by other researchers. In their results, they could include a comparison between the results of the LAMP and qPCR assays. Both were performed, if I understood correctly. Did any sample present different results between them? Please make this point clearer.

Reviewer #2: The study in novel. It provides insight into the transmission cycle of a devastating disease.

Reviewer #3: Abstract

- “Notably, the handling and consumption of snails sold at wet markets do not appear to increase the risk of eosinophilic meningitis in this region.” May this be because the freshwater snails selled in the wet market do not come from this area? Consider adding some info regarding this point in the discussion. Maybe Sulcospira species from the market are not positive because they come from other areas luckily free of cantonensis. Just an idea, but stating that it do not contribure… well, in this case. Has Sulcospira being reported as IH in other regions? If not, then it might not be a good IH, independently of being selled in the wet market.

Introduction

- L6: references mention only a few specific cases of the rat lungworm in wildlife. If not reporting all the species reported as accidental, I recomment to reduce the references to the two reviews (ref. 4 and 6). All animal groups mentioned in the rest of references appear in those two reference. In fact, reference 5 is included in the review of the 4…

- L13: Please, put the scientific names of the rat species found in Sumatra in brackets “… also identified in rats (Rattus tiomanicus…. and Rattus…).

- L15: add here the common name of Lissacatina fulica and delete it from line 19.

- L16: Add the S of the plural “South Sumatra ProvinceS”.

- L19: I would suggest unifying all the snail’s species photographs in a single figure containing the three photos at once. This way it is easier to follow and have a general idea on the species involved in the study.

- L22: “the abundance of these gastropods in Aceh province is high”. Please, add a reference for this affirmation.

- L26: As mentioned above, unify all snail photos in a single figure.

Materials and Methods

I am missing an important part here, the identification of the snails. Although I understand these species may not be difficult to identify, being this study focused on the snails, the correct identification of each genus mentioned is crucial and should be confirmed. Please, add the references used for the identification of the snails. If non used, consider performing some PCR of a few individuals per genus to be sure it is correct. Add a small section with all the methods.

- L62: Why was LAMP used and then qPCR instead of performing just the qPCR? Explain.

- L69: Why was the additional procedure of the emerging of Angiostrongylus performed in these snails instead of analysing just the tissue snips as in Lissachatina? Are we able to compare results using different methods?

- L75: Now the Sulcospira (also an aquatic snail) samples are analysed directly without doing the emergence of larvae step. Why is this? Why not performing the same analysis in all snails the same way? Results may differ because of the techniques. Please, clearly explain the reason to differ in the methodology per species.

- L80: I assume you wanted to say “99ºC”.

- L85: “with modification optimized for L3 larvae of A. cantonensis, when the pre-lyse phase was extended overnight”. I would rewrite it, as it seems not natural to me using that “when”. Maybe something like “extending the pre-lyse phase overnight as a modification for L3 larvae of A. cantonensis.

I would rewrite the whole Methods section; it is not clear if LAMP was performed in all samples or just in Lissachatina and Pomacea snails. Also, there is different methodology per genus without explanation.

Results

- L109: Please, give 95% Confidence Interval (exact method) together with the prevalence (add this in the section methods), and a bit on the descriptive analysis.

- L104: Please, delete the blank space between 6._7% (6.7%).

- L117: Fig 3 refers to a photograph of the Sulcospira snails. Do you mean the map of the Fig. 4? Please check. Also, important, ADD scale bar to maps.

- L119: I suggest writing “five DNA sequences varying from 322 to 554 bp from …”.

In this section I miss a clear result on the LAMP and posterior confirmation of the qPCR. Only final positives are reported, but it would be interesting to know if all LAMP positives were confirmed with qPCR or if there were some false positive pools.

Discussion

- L141: “Notably, both A. cantonensis and A. malaysiensis are zoonotic.”. Why this sentence here?

- L143: Are we sure on the absence of A. cantonensis in this species as we performed a different methodology?

- L160: I suggest adding these two references together with ref. 19: https://doi.org/10.1371/journal.pone.0094969 and https://doi.org/10.1016/j.onehlt.2023.100610.

- L166-168: Might this also be driven by rat distribution and densities?

- L173: Start the sentence with a “the”. The presence.

- Maybe a small sentence regarding why you do not differenciate between snail species in this study would be great. Consider adding something similar in the methods’ section.

References

- Ref. 29: Italics in scientific name.

Figures

- Figure 1-3: As mentioned above in the text, it would be clear to have the three snail photos in the same figure. Also, apart from the photo of L. fulica, the rest of the species are not clearly visible. Although these are good pictures of the snails in the environment and in the wet market, consider adding some photos with a clear view of the snail to see the morphology a bit more, especially when no info on the snails’ identification has been provided in the text.

- Figure 4: Add scale, at least in the zoomed map.

PLOS authors have the option to publish the peer review history of their article (what does this mean?). If published, this will include your full peer review and any attached files.

Reviewer #1: No

Reviewer #2: **Yes: **Anisuzzaman

Reviewer #3: No

**Figure resubmission:**
---

## [Decision Letter · Decision Letter 1]

22 Aug 2025

Dear Dr. Anettová Hidden Threats in Urban Environments: Angiostrongylus cantonensis in Banda Aceh’s Cityscape

PLOS Neglected Tropical DiseasesDear Dr. Anettová, Thank you for submitting your manuscript to PLOS Neglected Tropical Diseases. After careful consideration, we feel that it has merit but does not fully meet PLOS Neglected Tropical Diseases's publication criteria as it currently stands. Therefore, we invite you to submit a revised version of the manuscript that addresses the points raised during the review process. Please submit your revised manuscript within 30 days Sep 21 2025 11:59PM. If you will need more time than this to complete your revisions, please reply to this message or contact the journal office at plosntds@plos.org. Please include the following items when submitting your revised manuscript: * A rebuttal letter that responds to each point raised by the editor and reviewer(s). You should upload this letter as a separate file labeled 'Response to Reviewers'. This file does not need to include responses to any formatting updates and technical items listed in the 'Journal Requirements' section below. * A marked-up copy of your manuscript that highlights changes made to the original version. You should upload this as a separate file labeled 'Revised Manuscript with Track Changes'. * An unmarked version of your revised paper without tracked changes. You should upload this as a separate file labeled 'Manuscript'. If you would like to make changes to your financial disclosure, competing interests statement, or data availability statement, please make these updates within the submission form at the time of resubmission. Guidelines for resubmitting your figure files are available below the reviewer comments at the end of this letter. We look forward to receiving your revised manuscript. Kind regards, Alessandra Morassutti, PhDAcademic EditorPLOS Neglected Tropical Diseases Francesca TamarozziSection EditorPLOS Neglected Tropical Diseases

Shaden Kamhawi

co-Editor-in-Chief

Paul Brindley

co-Editor-in-Chief

**Journal Requirements:**

Please ensure that the CRediT author contributions listed for every co-author are completed accurately and in full.

At this stage, the following Authors/Authors require contributions: Lucia Anettová, Anna Šipková, Vojtech Baláž, Muhammad Hambal, Radovan Coufal, Jana Kačmaříková, Henni Vanda, Wahyu Eka Sari, and David Modrý. Please ensure that the full contributions of each author are acknowledged in the "Add/Edit/Remove Authors" section of our submission form.

  **Reviewers' comments:** Reviewer's Responses to Questions

**Key Review Criteria Required for Acceptance?**

**Methods:**

-Are the objectives of the study clearly articulated with a clear testable hypothesis stated?

-Is the study design appropriate to address the stated objectives?

-Is the population clearly described and appropriate for the hypothesis being tested?

-Is the sample size sufficient to ensure adequate power to address the hypothesis being tested?

-Were correct statistical analysis used to support conclusions?

-Are there concerns about ethical or regulatory requirements being met?

Reviewer #1: All previous recommendations have been met. The manuscript is better presented.

Reviewer #2: PCR methods are still cryptic. Please provide the detail of the provide the primer sequences of ITS1 with thermal cycle of the ITS1 based PCR. The method must be reproducible. You provided the reference. If any one want to follow your method then he/she need to check the cross ref. Please make the story self-dependent. Otherwise, the manuscript is fine and interesting.

Reviewer #3: Methods section has now improved a lot. I would like to thank the authors for clarifying all my concerns related to the identification of the snails and the methodology used in the analysis. Now is clearly stated why using a different methodology depending on the snails.

I would like to add some minor comments:

- Line 85: I would move the methods regarding to the prevalence calculation and descriptive analysis to the end of the methods section, as we need to know first how do you analyse the snails (LAMP and qPCR methodology), and based on those results, calculate the prevalence.

- Line 74: You mention here that analysis was performed in pools of 10 individuals for Sulcospira, but no info on pools of the other species was reported here, but it is explained in the LAMP section. As pools were used only for LAMP, I would move all info about pools to the LAMP section, mentioning how many individuals were mixed in pools and why the number differed between species (I suppose it was because of the size of the snail, but it must be mentioned in the text).

-Line 106: This is the methods section, so we do not know yet that only A. fulica was positive. Rewrite it in a more general way like: Pools that tested positive to LAMP analysis underwent the species specific qPCR using isolated DNA from each individual included in the pool....

**Results:**

-Does the analysis presented match the analysis plan?

-Are the results clearly and completely presented?

-Are the figures (Tables, Images) of sufficient quality for clarity?

Reviewer #1: All previous recommendations have been met. The manuscript is much better presented.

Reviewer #2: Well done

Reviewer #3: Thank you for answering all my previous comments, now the article gained in clarity and accuracy.

**Conclusions:**

-Are the conclusions supported by the data presented?

-Are the limitations of analysis clearly described?

-Do the authors discuss how these data can be helpful to advance our understanding of the topic under study?

-Is public health relevance addressed?

Reviewer #1: The findings are relevant and well presented

Reviewer #2: Based on solid proof

Reviewer #3: Yes

**Editorial and Data Presentation Modifications?**

Reviewer #1: I just leave a suggestion that the authors add a reference regarding the identification of Pomacea sp.

Reviewer #2: (No Response)

Reviewer #3: (No Response)

**Summary and General Comments:**

Reviewer #1: (No Response)

Reviewer #2: (No Response)

Reviewer #3: I think the manuscript has improved a lot and should be accepted for publication after addressing those three minor comments.

PLOS authors have the option to publish the peer review history of their article (what does this mean?). If published, this will include your full peer review and any attached files.

Reviewer #1: **Yes: **Jucicleide Ramos-de-Souza

Reviewer #2: **Yes: **Anisuzzaman

Reviewer #3: No

**Figure resubmission:** While revising your submission, we strongly recommend that you use PLOS’s NAAS tool (https://ngplosjournals.pagemajik.ai/artanalysis) to test your figure files. NAAS can convert your figure files to the TIFF file type and meet basic requirements (such as print size, resolution), or provide you with a report on issues that do not meet our requirements and that NAAS cannot fix.

After uploading your figures to PLOS’s NAAS tool - https://ngplosjournals.pagemajik.ai/artanalysis, NAAS will process the files provided and display the results in the "Uploaded Files" section of the page as the processing is complete. If the uploaded figures meet our requirements (or NAAS is able to fix the files to meet our requirements), the figure will be marked as "fixed" above. If NAAS is unable to fix the files, a red "failed" label will appear above. When NAAS has confirmed that the figure files meet our requirements, please download the file via the download option, and include these NAAS processed figure files when submitting your revised manuscript.**Reproducibility:**

---

## [Decision Letter · Decision Letter 2]

11 Oct 2025

Dear Dr. Anettová,

We are pleased to inform you that your manuscript 'Hidden Threats in Urban Environments: Angiostrongylus cantonensis in Banda Aceh’s Cityscape' has been provisionally accepted for publication in PLOS Neglected Tropical Diseases.

Best regards,

Alessandra Morassutti, PhD

Academic Editor

Francesca Tamarozzi

Section Editor

Shaden Kamhawi

co-Editor-in-Chief

Paul Brindley

co-Editor-in-Chief

Reviewer's Responses to Questions

**Key Review Criteria Required for Acceptance?**

**Methods**

-Are the objectives of the study clearly articulated with a clear testable hypothesis stated?

-Is the study design appropriate to address the stated objectives?

-Is the population clearly described and appropriate for the hypothesis being tested?

-Is the sample size sufficient to ensure adequate power to address the hypothesis being tested?

-Were correct statistical analysis used to support conclusions?

-Are there concerns about ethical or regulatory requirements being met?

Reviewer #2: Well described

Reviewer #3: (No Response)

**Results**

-Does the analysis presented match the analysis plan?

-Are the results clearly and completely presented?

-Are the figures (Tables, Images) of sufficient quality for clarity?

Reviewer #2: Well written

Reviewer #3: (No Response)

**Conclusions**

-Are the conclusions supported by the data presented?

-Are the limitations of analysis clearly described?

-Do the authors discuss how these data can be helpful to advance our understanding of the topic under study?

-Is public health relevance addressed?

Reviewer #2: Based on solid evidence

Reviewer #3: (No Response)

**Editorial and Data Presentation Modifications?**

Reviewer #2: Accept

Reviewer #3: Thank you for taking into consideration my previous comments.

My last suggestion is related to the title of the manuscript. I would suggest adding the country or the province and country in the title afer Cityscape, as not all readers would know where has the study performed by reading "Banda Aceh's". So I would say something similar to: "Hidden Threats in Urban Environments: Angiostrongylus cantonensis in Banda Aceh’s Cityscape (Sumatra, Indonesia)".

After this, I have no other comments regarding the manuscript, it should be accepted for publication as it is with that minor modification of the title, if the editor agrees.

Good job!

**Summary and General Comments**

Reviewer #2: All comments and suggestions have been addressed

Reviewer #3: (No Response)

PLOS authors have the option to publish the peer review history of their article (what does this mean?). If published, this will include your full peer review and any attached files.

Reviewer #2: **Yes: **Anisuzzaman

Reviewer #3: No

---

## [Editor Report · Acceptance letter]

Dear Dr. Anettová,

We are delighted to inform you that your manuscript, "Hidden Threats in Urban Environments: Angiostrongylus cantonensis in Banda Aceh’s Cityscape," has been formally accepted for publication in PLOS Neglected Tropical Diseases.

Best regards,

Shaden Kamhawi

co-Editor-in-Chief

Paul Brindley

co-Editor-in-Chief
